# Anxiolytic-like Effects and Quantitative EEG Profile of Palmitone Induces Responses Like Buspirone Rather Than Diazepam as Clinical Drugs

**DOI:** 10.3390/molecules28093680

**Published:** 2023-04-24

**Authors:** Daniela Onofre-Campos, María Eva González-Trujano, Gabriel Fernando Moreno-Pérez, Fernando Narváez-González, José David González-Gómez, Benjamín Villasana-Salazar, David Martínez-Vargas

**Affiliations:** 1Laboratorio de Neurofarmacología de Productos Naturales, Dirección de Investigaciones en Neurociencias, Instituto Nacional de Psiquiatría Ramón de la Fuente Muñiz, Calz. México-Xochimilco 101, Col. San Lorenzo Huipulco, Tlalpan, Ciudad de México 14370, Mexico; 2Biología Experimental, Universidad Autónoma Metropolitana, Ciudad de México 09340, Mexico; 3ISSSTE Hospital Regional General Ignacio Zaragoza, Calz. Ignacio Zaragoza 1840, Juan Escutia, Iztapalapa, Ciudad de México 09100, Mexico; 4Laboratorio de Neurofisiología del Control y la Regulación, Dirección de Investigaciones en Neurociencias, Instituto Nacional de Psiquiatría Ramón de la Fuente Muñiz, Calz. México-Xochimilco 101, Col. San Lorenzo Huipulco, Tlalpan, Ciudad de México 14370, Mexico

**Keywords:** anxiety, delta band, electroencephalography, fatty acids, 16-hentriacontanone, palmitone

## Abstract

Anxiety is a mental disorder with a growing worldwide incidence due to the SARS-CoV-2 virus pandemic. Pharmacological therapy includes drugs such as benzodiazepines (BDZs) or azapirones like buspirone (BUSP) or analogs, which unfortunately produce severe adverse effects or no immediate response, respectively. Medicinal plants or their bioactive metabolites are a shared global alternative to treat anxiety. Palmitone is one active compound isolated from *Annona* species due to its tranquilizing activity. However, its influence on neural activity and possible mechanism of action are unknown. In this study, an electroencephalographic (EEG) spectral power analysis was used to corroborate its depressant activity in comparison with the anxiolytic-like effects of reference drugs such as diazepam (DZP, 1 mg/kg) and BUSP (4 mg/kg) or 8-OH-DPAT (1 mg/kg), alone or in the presence of the GABA_A_ (picrotoxin, PTX, 1 mg/kg) or serotonin 5-HT_1A_ receptor antagonists (WAY100634, WAY, 1 mg/kg). The anxiolytic-like activity was assayed using the behavioral response of mice employing open-field, hole-board, and plus-maze tests. EEG activity was registered in both the frontal and parietal cortex, performing a 10 min baseline and 30 min recording after the treatments. As a result, anxiety-like behavior was significantly decreased in mice administered with palmitone, DZP, BUSP, or 8-OH-DPAT. The effect of palmitone was equivalent to that produced by 5-HT_1A_ receptor agonists but 50% less effective than DZP. The presence of PTX and WAY prevented the anxiolytic-like response of DZP and 8-OH-DPAT, respectively. Whereas only the antagonist of the 5-HT_1A_ receptor (WAY) inhibited the palmitone effects. Palmitone and BUSP exhibited similar changes in the relative power bands after the spectral power analysis. This response was different to the changes induced by DZP. In conclusion, brain electrical activity was associated with the anxiolytic-like effects of palmitone implying a serotoninergic rather than a GABAergic mechanism of action.

## 1. Introduction

Anxiety is one of the most common psychiatric conditions worldwide, with an estimated lifetime prevalence of 3.8–25% among adults [1]. Currently, and in the face of the SARS-CoV-2 virus (COVID-19) pandemic, this prevalence has increased due to this unexpected crisis period. An international study carried out in 2020 in 234 countries and published by The Lancet estimates that 26% of the cases of anxiety disorders in the world have threatened the pandemic. This means that in 2020, 76 million new cases of anxiety were detected compared to those expected, where women remain with a greater increase in prevalence than men [2].

Anxiety disorders generally have a common treatment, both psychological and pharmacological. However, pharmacotherapeutic strategies for treating anxiety are not always effective and are associated with several adverse effects [3,4]. In the case of benzodiazepines (BDZs), as first-line pharmacological treatment and commonly used due to their rapid onset of action, they can lead to a moderate improvement of symptoms and central nervous system (CNS) depression, leading to dizziness, fatigue, increased reaction time, and deterioration of cognitive functions, as well as tolerability and dependence [3,5,6]. In the case of azapirones (serotonin 5-HT_1A_ receptor partial agonists) such as buspirone (BUSP) and analogs, they are associated with less drowsiness, psychomotor impairment, alcohol potentiation, addiction, or abuse potential than BDZs, and seemed to be well tolerated, but they may require at least more than a week to establish their anxiolytic efficacy [7].

Novel therapeutic strategies are continuously needed. A meta-analysis reported that polyunsaturated fatty acids, such as Omega-3, produced anxiolytic effects in humans [8]. Similarly, preclinical studies had shown reduced anxiety-like behavior when rodents were treated with a mixture of long-chain saturated fatty acids [9,10,11]. Regarding medicinal plants and their bioactive metabolites, a non-polar fraction of *Tilia americana* containing the terpene beta-sitosterol and a mixture of fatty acids produced anxiolytic-like effects in rodents [12]. In agreement with this report, a non-polar fraction of *Bertholletia excelsa* seeds, highly enriched with oleic acid (59.97%) and palmitic acid (21.42%), also produced significant anxiolytic-like effects in mice [10]. All these data together suggest that natural products are a source of several fatty acids with possible anxiolytic properties [13]. 

Palmitone (16-hentriacontanone) has been used to produce lipidized nucleosides as potential brain markers with the ability to cross the blood–brain barrier as a single photon emission computed tomography (SPECT) agent for neuroimaging studies to improve early diagnostics [14] supporting its easy pass through the brain tissue. In the brain, palmitone produces depressant activity reducing excitability like paroxysms and protecting cerebral regions such as the hippocampus from neuronal damage, not only in adult rats but also in prepubertal rats [15,16]. Concerning pain relief, palmitone reduced nociceptive response in abdominal, nociceptive, and inflammatory tests in rodents involving opioid and 5-HT_1A_ receptors [17]. It has been found to be a major component representing up to 48% of the total cuticular wax of *A. squamosa* which antimicrobial activity against selected Gram-positive and Gram-negative bacterial and fungal strains have been reported to be significantly higher or equivalent in the presence of palmitone, respectively, in comparison to some isomeric hydroxy ketones [18]. All these studies together support the biological relevance of palmitone on the CNS and in other cases.

In a preliminary study, we reported the depressant effects of palmitone, an aliphatic ketone identified and extracted from a hexane extract of *Annona diversifolia* Saff. leaves by demonstrating anxiolytic-like effects in mice [19]. Interestingly, the anxiolytic-like results of palmitone differed from those of diazepam (DZP) since this ketone did not alter motor responses as DZP did [19]. López-Rubalcava et al. (2006) reported that a non-polar extract of *Annona cherimolia* leaves produced anxiolytic-like effects in mice due to the presence of beta-sitosterol and palmitone, suggesting a possible involvement of GABA_A_/benzodiazepines (BDZs) receptors in the antianxiety effects of palmitone [20]. However, the precise mechanism of action of this aliphatic ketone to explain its anxiolytic properties has not been explored enough. Preliminary studies reporting the biological effects of isolated palmitone from the hexane extracts obtained from *A. diversifolia* and *A. muricata* on the CNS, such as anticonvulsant and anxiolytic [19,20,21] suggested a possible mechanism of action through GABAergic neurotransmission. Another study reported that opioids and serotonin 5-HT_1A_ receptors were involved in the antinociceptive effects of *A. diversifolia* [17]. However, direct exploration of this pure compound alone and in the presence of selective antagonists was not previously investigated for the antianxiety response. To clarify a possible mechanism of action for palmitone in preclinical anxiety, the participation of serotonin 5-HT_1A_ receptor but not the GABA/BDZs site receptor was corroborated in this study by using selective antagonists alone or combined with palmitone. Our present results using not only behavior but also EEG analysis reinforced that this aliphatic ketone differed from that observed for BDZ such as diazepam.

Electroencephalograms (EEG) and spectral power analyses are widely used tools that provide deeper insight into CNS illnesses like anxiety [22]. Several studies have suggested that anxiety disorders have distinctive EEG activity patterns in frontal cortical areas during resting state in humans, specifically in the alpha frequency band [23,24]. In fact, EEG has been commonly used as a screening measurement of the efficacy and safeness not only of clinical drugs but also for new treatments, such as medicinal plants with CNS activity [21,25,26,27]. To know the influence of palmitone in the CNS and its possible antianxiety mechanism of action, cortical EEG activity was investigated in this study in mice receiving doses producing anxiolytic-like responses compared to those observed in clinical anxiolytic drugs DZP and BUSP, by exploring their quantitative EEG profile and participation of GABA_A_ and serotonin 5-HT_1A_ receptors using antagonists.

## 2. Results

### 2.1. Anxiolytic-like Effects of Palmitone and Clinical Anxiolytics Drugs

A significant reduction in the ambulatory and head-dipping behaviors was observed in mice receiving palmitone (30 mg/kg, i.p.). These responses were equivalent to that presented in the groups treated with DZP (1 mg/kg, i.p.) or BUSP (4 mg/kg, i.p.) as observed in the open-field (Figure 1A) (F_3, 20_ = 18.14, *p* < 0.0001) and hole-board (Figure 1B) (F_3, 20_ = 27.97, *p* < 0.0001) tests. Whereas a significant increase was obtained in the time that mice spent in open-sided arms of the plus-maze test in all the treatments, but the response of palmitone (*p* < 0.001) was almost half of that obtained with the anxiolytic drugs DZP and BUSP (*p* < 0.0001), in the plus-maze test (F_3, 20_ = 43.83, *p* < 0.0001) (Figure 1C). 

### 2.2. GABA_A_ Receptors Involvement in the Anxiolytic-like Effects of Palmitone 

The presence of PTX (1 mg/kg, i.p.) did not produce per se changes in the ambulatory, head-dipping, and time spent in open-sided arms behaviors in mice, as compared to the vehicle group. In contrast, the antagonist completely prevented the significant anxiolytic-like responses of DZP in the open field (F_5, 30_ = 12.53, *p* < 0.0001), hole-board (F_5, 30_ = 3.808, *p* < 0.0001), and plus-maze (F_5, 30_ = 29.58, *p* < 0.0001) tests (Figure 2A–C). In addition, the significant anxiolytic-like response of palmitone remained unchanged in the presence of PTX (Figure 2A–C).

### 2.3. Serotonin 5-HT_1A_ Receptors Involvement in the Anxiolytic-like Effects of Palmitone 

Pretreatment with WAY (1 mg/kg, i.p.) alone did not produce per se changes in the ambulatory and head-dipping behaviors in mice, compared to the vehicle group (Figure 3A and Figure 3B, respectively). A significant response was obtained when the serotonin 5-HT_1A_ receptor antagonist was evaluated in the plus-maze test (Figure 3C) (F_5, 30_ = 12.49, *p* < 0.0001). This antagonist completely prevented the significant anxiolytic-like responses of a serotonin 5-HT_1A_ receptor agonist, 8-OH-DPAT (1 mg/kg, s.c.), in the open-field and hole-board tests (F_5, 30_ = 18.61, *p* < 0.0001) (Figure 3A,B), with a partial reduction when it was evaluated in the plus-maze test (F_5, 30_ = 12.49, *p* < 0.0001) (Figure 3C). Regarding the significant anxiolytic-like effect of palmitone (30 mg/kg, i.p.), this remained without alteration in the presence of the antagonist in the open-field and hole-board tests, but complete prevention was produced in the time spent by mice in the open-sided arms in the plus-maze test (F_5, 30_ = 12.49, *p* < 0.0001) (Figure 3C).

### 2.4. EEG Patterns and Spectral Power Analysis

#### 2.4.1. EEG Changes after Treatment of Palmitone and Anxiolytics Drugs

Thirty minutes after treatments, the following EEG changes were observed in the frontal (FC) and parietal (PC) cortex. Mice treated with vehicle did not show noticeable EEG activity resembling the high-frequency/low-amplitude EEG pattern observed in the baseline (EEG activity desynchronized at frequencies from 4–8 Hz) (Figure 4A). Both palmitone (30 mg/kg) and BUSP (4 mg/kg) caused a slowdown in the EEG activity, characterized by a low-frequency/high-voltage pattern in the 1–6 Hz frequency range (Figure 4B and Figure 4C, respectively). In contrast, DZP (1 mg/kg) provoked an acceleration in the EEG characterized by an increase in high-frequency/low-voltage activity, which occurred at times in bursts ~8–15 Hz (Figure 4D).

#### 2.4.2. EEG Spectral Power Analysis after Palmitone and Anxiolytic Drugs

The spectral power analysis showed a relationship between drug-treatment response, frequency band, and cortical region compared to baseline activity (Figure 5). The one-sample *t*-test showed that in the frontal and parietal cortex, the vehicle did not cause any significant change in the analyzed frequency bands (Figure 5A). DZP (1 mg/kg, i.p.) provoked significant decreases in the delta band and increases in the beta band in both regions and hemispheres ([delta] left FC: *t*_(5)_ = −3.38, *p* = 0.011; right FC: *t*_(5)_ = −8.46, *p* = 0.0004; left PC: *t*_(5)_ = −7.97, *p* = 0.0005; right PC: *t*_(5)_ = −9.83, *p* = 0.0002; [beta] left FC: *t*_(5)_ = 6.96, *p* = 0.0009; right FC: *t*_(5)_ = 4.19, *p* = 0.0085; left PC: *t*_(5)_ = 7.44, *p* = 0.0007; right PC: *t*_(5)_ = 7.85, *p* = 0.0005) (Figure 5B). In contrast, BUSP (4 mg/kg, i.p.) induced significant increases in the delta band and decreases in the alpha band in both cortical regions bilaterally ([delta] left FC: *t*_(5)_ = 3.34, *p* = 0.020; right FC: *t*_(5)_ = 3.98, *p* = 0.010; left PC: *t*_(5)_ = 5.04, *p* = 0.0039; right PC: *t*_(5)_ = 3.04, *p* = 0.028; [alpha] left FC: *t*_(5)_= −2.96, *p* = 0.037; right FC: *t*_(5)_ = −3.26, *p* = 0.022; left PC: *t*_(5)_ = −3.16, *p* = 0.025; right PC: *t*_(5)_ = −2.86, *p* = 0.035) (Figure 5C). Regarding palmitone (30 mg/kg, i.p.), it provoked increases in the delta band and decreases in the alpha band in the four analyzed regions ([delta] left FC: *t*_(5)_ = 3.98, *p* = 0.010; right FC: *t*_(5)_ = 3.34, *p* = 0.020; left PC: *t*_(5)_ = 3.29, *p* = 0.021; right PC: *t*_(5)_ = 3.47, *p* = 0.017; [alpha] left FC: *t*_(5)_ = −2.90, *p* = 0.033; right FC: *t*_(5)_ = −2.08, *p* = 0.037; left PC: *t*_(5)_ = −5.09, *p* = 0.003; right PC: *t*_(5)_ = −6.24, *p* = 0.0015) (Figure 5D). 

Comparing the relative power among groups shows significant differences between drug-treatment responses in frequency bands’ relative power (Figure 4). The one-way ANOVA test revealed significant differences in the delta (left FC: F_3, 20_ = 10.56, *p* < 0.0002; right FC: F_3, 20_ = 11.48, *p* < 0.0001; left PC: F_3, 20_ = 11.58, *p* < 0.0001; right PC: F_3, 20_ = 8.32, *p* < 0.003), alpha (left FC: F_3, 20_ = 3.9, *p* < 0.05; right FC: F_3, 20_ = 5.23, *p* < 0.01), and the beta bands (left FC: F_3, 20_ = 15.04, *p* < 0.001; right FC: F_3, 20_ = 11.05, *p* < 0.0002; left PC: F_3, 20_ = 12.52, *p* < 0.0001; right PC: F_3, 20_ = 16.37, *p* < 0.0001) (Figure 6A–D).) The *post hoc* analysis indicated that palmitone and BUSP exhibited a higher proportion of the delta frequency band in both the frontal (*p* < 0.001, Figure 6A,B) and parietal cortex (*p* < 0.05, Figure 6C,D) compared to DZP. Palmitone provoked a diminution in the alpha band compared to DZP in the frontal cortex (*p* < 0.05) (Figure 6A,B). In contrast, DZP induced an increase in the beta band in comparison to BUSP (*p* < 0.001), palmitone (*p* < 0.001), and vehicle (*p* < 0.05) in the four analyzed regions (*p* < 0.001) (Figure 6A–D, right panels). Additionally, DZP provoked a significant diminution of the delta band in both areas and hemispheres compared to the vehicle (*p* < 0.05, Figure 6A–D). 

#### 2.4.3. EEG Spectral Power Analysis after Palmitone Compared with Agonist and Antagonist of GABA_A_ and 5-HT_1A_ Receptors

It is important to mention that changes in EEG spectral power from the left hemisphere (Figure 7 and Figure 8) were equivalent to those observed in the right hemisphere on both frontal and parietal regions. In order to not duplicate results, data obtained from the right frontal and parietal cortex from EEG are reported in the main text, whereas the EEG analysis dataset from the left hemisphere can be found in Appendix A. Firstly, the inhibition of GABA_A_ receptors in the presence of the antagonist PTX (1 mg/kg, i.p.) was examined, which provoked a significant diminution in the beta band with respect to baseline per se in the parietal cortex (*t*_(5)_ = −4.41, *p* = 0.006) (Figure 7B). However, this antagonist did not modify the observed changes in the relative power induced by palmitone (30 mg/kg, i.p.) (Figure 7). In contrast, PTX (1 mg/kg, i.p.) significantly prevented the EEG changes obtained with DZP (1 mg/kg, i.p.) in the beta band in both regions (right FC: F_5, 30_ = 26.09, *p* < 0.0001; right PC: F_5, 30_ = 24.22, *p* < 0.0001) (Figure 7A,B).

Regarding the serotoninergic neurotransmission, it was examined whether activation of 5-HT_1A_ receptors in the presence of the 8-OH-DPAT agonist (1 mg/kg, s.c.), or inhibition with the antagonist WAY (1 mg/kg, i.p) affects the relative power of the bands in the frontal and parietal cortex. As a result, the baseline compared with the response of 8-OH-DPAT (1 mg/kg, s.c.) elicited a significant increase in the delta band in the parietal cortex (*t*_(5)_ = −4.29, *p* = 0.007). Whereas a significant increase was observed in the gamma band (right FC: *t*_(5)_ = 3.26, *p* = 0.022; right PC: *t*_(5)_ = 6.48, *p* = 0.0013) and a significant decrease in the alpha band (right FC: *t*_(5)_ = −8.77, *p* = 0.0003; right PC: *t*_(5)_ = −6.43, *p* = 0.0013) in both cerebral cortices (Figure 8). In the case of WAY (1 mg/kg, i.p.) alone, it provoked a decrease in the beta band in the parietal cortex (*t*_(5)_ = −5.05, *p* = 0.003, Figure 8B). Regarding the WAY + palmitone group, it showed an increase in the delta band and a decrease in the alpha band in both cerebral cortices ([delta] right FC: *t*_(5)_ = 4.50, *p* = 0.006; right PC: *t*_(5)_= 4.29, *p* = 0.007; [alpha] right FC: *t*_(5)_ = −2.08, *p* = 0.037; right PC: *t*_(5)_ = −7.37, *p* = 0.0007) (Figure 8A,B). In addition, the effects in the alpha band induced by 8-OH-DPAT (1 mg/kg, s.c.) were prevented by WAY in the right frontal and parietal cortex (right FC: F_5, 30_ = 5.61, *p* < 0.001; right PC: F_5, 30_ = 57.62, *p* < 0.0001) (Figure 8A,B). The effects of palmitone in the presence of WAY remained without changes in the relative power bands.

#### 2.4.4. Docking Analysis

A molecular docking approach for target validation was conducted to support palmitone’s drug–receptor relationship and its anxiolytic-like effects observed in the in situ and in vivo protocol. The results showed that palmitone possesses binding energy in the range of −4.0 to −3.6 kcal/mol for a possible interaction on the serotonin 5-HT_1A_ receptor (Figure 9A) in comparison to a −5.1 to −4.8 kcal/mol for the GABA_A_ receptor (Figure 9B) suggesting low drug ability between palmitone and one of these two receptors as the mechanism of action for their anxiolytic-like response observed in the behavior (Figure 1, Figure 2 and Figure 3) and in the EEG analysis (Figure 4, Figure 5, Figure 6, Figure 7 and Figure 8).

The molecular docking of palmitone interaction was compared to those obtained for the reference drugs. Regarding BUSP and 8-OH-DPAT agonists of the serotonin 5-HT_1A_ receptor, their binding energy was calculated in the range of −7.7 to −6.7 compared to −6.6 to −5.7, respectively. The corresponding steric interactions are observed in Figure 10. Concerning diazepam interaction on the GABA_A_ receptor, a range of binding energy −8.3 to −7.1 was calculated (Figure 11).

## 3. Discussion

In this study, the alterations in the EEG activity bands of the frontal and parietal cortex of mice induced by the presence of palmitone, a natural product with anxiolytic properties, were investigated and compared with the EEG profile of the clinical anxiolytics BUSP and DZP. Additionally, the behavioral anxiolytic-like effects and the EEG activity were analyzed in combination with chemical antagonists to know the role of GABA_A_/BDZs site and serotonin 5-HT_1A_ receptor as possible mechanisms of action.

The presence of palmitone as an abundant secondary metabolite has been found in several plants such as in the hexane extract of leaves of *Aristolochia cordigera* [28] or in the *Annona* species [19,20], but also in the ethanol extracts [21]. For example, the anxiolytic-like effects of a methanol extract of the *Aristolochia indica* leaves was reported using light/dark test [29]. Several fatty acids, such as palmitoleic acid (0.14%), palmitic acid (21.42%), linoleic acid (11.02%), oleic acid (59.97%), and stearic acid (7.44%), were identified as constituents of an hexanic extract of the *Bertholletia excelsa* seeds producing significant anxiolytic-like effects [10]. A significant reduction in the exploratory activity of mice assessed in the open-field and hole-board tests, as well as an increase in the time spent in open-side arms of the plus maze test, were observed as palmitone’s anxiolytic-like effects when evaluations were carried out thirty minutes after treatment administration in the present study. These results agreed with a preliminary study supporting the pharmacological evidence of the depressant effects of palmitone [19], which were compared to those obtained with DZP and BUSP, two reference drugs commonly used for anxiety therapy. It has been reported that non-polar fatty acids constituents like palmitic acid might extend the anxiolytic-like effects of some agents, partly due to an improvement in brain penetration, such as in the case of the potential anxiolytic neuropeptide limited by its inability to cross the blood–brain barrier [30]. It might occur by lipidization, which means appending the lipid chain of ketone fatty acid, palmitone (16-hentriacontanone), used to synthesize and enhance liposolubility of agents to improve their capability of penetrating the blood–brain barrier [14]. Palmitone is an aliphatic ketone with non-polar characteristics equivalent to palmitic acid. Concentrations of individual fatty acids might also be associated with significant changes in CNS, including sedative effects or even anxiety-like behavior, as it was observed 24 h after administration of myristic acid and palmitic acid in mice, which reduced locomotion and exploration in a zero-maze but not in Y-maze or the swimming test. This effect was associated with serotonin neurotransmission since the 5-HT_1A_ metabolite was increased by 33% in the amygdala 24 h after palmitic acid treatment [31]. Fatty acids are also precursors of prostaglandins such as PGE2, and to a lesser degree, PGE1, which are reported to inhibit the hormones released under stress, mainly catecholamines but also histamine, gastrin, and serotonin [32,33]. Anxiolytic-like effects of fatty acids (C_6_–C_18_) have also been associated with GABAergic neurotransmission in rats [11]. It is noteworthy that therapeutic responses for neurodegenerative diseases require repetitive administration of clinical drugs. Since most people with anxiety disorders use clinical drugs very often, sometimes daily, it is important for future studies to consider the antianxiety effects of palmitone after chronic administration, as already reported for its anticonvulsant properties after repeated treatment in the electrical amygdala kindling in rats without observing the presence of toxicity [34].

The precise mechanism of palmitone in producing its anxiolytic effects remains unknown. In this study, the role of two inhibitory receptors already known as implicated in the anxiolytic effects of clinical drugs commonly used in anxiety therapy (DZP and BUSP), such as GABA_A_/BDZs site and serotonin 5-HT_1A_ receptors, respectively, was explored. Anxiolytic-like effect of *A. cherimolia* was previously reported due to the presence of palmitone as one bioactive metabolite and a GABA/BDZs receptor modulation [20]. As previously mentioned, GABA is the main neurotransmitter system influenced by potential anxiolytic drugs such as BDZs [32,35]. Anxiolytic-like response of several drugs modulating GABA_A_ receptor has been blocked in the presence of PTX but not by flumazenil or bicuculine, suggesting that GABA_A_ modulation is more efficiently prevented by obstruction of the chloride channel [9,11,36]. Despite palmitone at 30 mg/kg, i.p. produced depressant effects equivalent to those of DZP (1 mg/kg, i.p.) and BUSP (4 mg/kg, i.p.), its behavioral response was not inhibited in the presence of the GABA antagonist PTX, as for DZP, proposing that a GABAergic mechanism is not the primary influence in its anxiolytic-like effects. In contrast, the significant anxiolytic-like response of palmitone obtained in the time spent by mice in open-sided arms of the plus-maze test was wholly prevented in the presence of the antagonist of the serotonin 5-HT_1A_ receptor, WAY100635 (1 mg/kg, i.p.) suggesting that this receptor is involved in the depressant activity of palmitone. 

Brain electrical activity is characterized by the amplitude or strength of the oscillations. Furthermore, desynchronization is considered because an increase in these oscillations refers to greater amplitude and power of brain electrical activity during the central depressant activity. Therefore, EEG analysis is a quantitative method to assess brain activity in pathological conditions such as anxiety and in the anxiolytic therapy of drugs. As observed in the results of this study, the EEG bands were analyzed by recording the frontal and parietal regions of mice receiving one of two clinical anxiolytic drugs or palmitone. All these three treatments significantly modified the basal activity observed in the vehicle group, suggesting a CNS depressant activity due to their anxiolytic-like effect. The treatment with palmitone slowed the complete EEG pattern in mice’s right and left parietal and frontal cortex. This effect was equivalent to those produced with the anxiolytic drug BUSP but contrasted to that obtained in the presence of DZP. These data agree with the differences observed in the behavioral responses. 

It is known that EEG exhibits a broad spectrum of oscillation frequencies that can be classified into different frequency bands such as delta (0.5–4 Hz), theta (4–8 Hz), alpha (8–12 Hz), beta (12–30 Hz), and gamma (3–80 Hz) [37]. These bands can be better characterized depending on the brain area associated with different neural processes [38]. For example, it has been reported that brain structures such as the frontal cortex and the hippocampus are involved in depression and anxiety [39]. Therefore, they are considered essential brain areas where a mixture of fatty acids plays an important role when exercising. Moreover, physical activity impacts brain function associated with reduced anxiety, partly due to changes in the brain fatty acids profile such as myristic, stearic, and palmitic acid production stimulated in running mice [40]. To this respect, a metabolomic analysis described remarkable differences mainly in the heptadecanoic, stearic, and palmitic acid (hexadecanoic acid) levels due to physical activity associated with a significant anxiolytic-like effect evaluated in the plus-maze and hole-board tests in mice [40]. The behavioral anxiolytic-like response of agonists of both inhibitory receptors explored in this study was inhibited in the respective antagonists’ presence in all the experimental models. Nevertheless, the anxiolytic-like response of palmitone depended on the anxiety test explored, suggesting that different mechanisms of DZP and BUSP are involved in the case of this metabolite to produce its depressant activity. 

In this study, alterations in particular band power in the EEG profile observed in the right and left frontal and parietal cortex were quantified. Characterization of herbal extracts targeting the CNS remains a continuous challenge to pharmacology. Usually, several different animal tests must be passed to determine in which direction a particular extract might act. Due to several molecular entities or ingredients with possibly other mechanisms of action, linear dose–response relationships cannot always be expected. The oscillatory patterns of EEG are ending physiological events driven by underlying mechanisms produced by the net effect of the drug’s action on multiple neurotransmitter systems and achieve information on local neuronal and synaptic activity [41]. In our study, EEG and spectral power analysis provided profound advantages to investigate the net effects of the treatments and distinguish distinctive EEG spectral profiles from the anxiolytics BUSP, DZP, and palmitone. The efficacy of BDZs for anxiety and mood disorders is well-known but not without adverse effects limiting their use [42,43]. Their CNS activities depend on their mechanism of action, such as an allosteric interaction on the GABA_A_ receptors, mainly those on the cortical brain regions [44], where an EEG analysis has reported an influence on the beta band demonstrating an increase, not only in experimental animals [42,45,46,47] but also in humans [48,49]. This effect has been proposed as a quantitative biomarker for GABA_A_ receptor modulation [50] and has been corroborated in the presence of clorazepate in patients with generalized anxiety disorder observed in the central and parietal cortex [51]. These results agree with human and rodent literature indicating that the effect of drugs such as BDZs on the brain electric activity is equivalent in central and parietal regions [41,45,52,53] and emphasize the importance of multi-lead recording in quantitative analysis of the EEG in order to document the effects on the brain electrical activity in the presence of drugs or natural products.

Interestingly, the spectral signature of palmitone was significantly different from DZP. At the same time, its spectral signature was equivalent to BUSP in the delta and alpha bands but without difference in the other ones. An increase in the slow frequency of bands is known to be associated with tranquilizing effects depending mainly on the thalamocortical oscillation [54]. In contrast, gamma oscillations reflected cortical hyperexcitability of stressed animals [55]. EEG changes obtained with DZP or 8-OH-DPAT were prevented in the presence of the antagonists PTX or WAY, respectively, but these did not modify the effects provoked by palmitone. It has been reported that delta activity is modulated by acetylcholine [56], whereas the alpha band reflects changes in serotoninergic [57] and dopaminergic transmission [58]. As experimentally demonstrated in the present study, palmitone provoked an increase in the delta band and a decrease in the alpha band, suggesting the inactivation of the cholinergic system and probably the activation of the serotoninergic and dopaminergic systems. The similitude of the EEG pattern of palmitone and BUSP might be related to a non-sedative anxiolytic activity conversely to the profile and the anxiolytic and sedative effects recognized for DZP [43,47], which also produced brain activity in the presence of spindles. BUSP is one of the current clinically used azapirones; as a partial serotonin 5-HT_1A_ agonist produces low abuse potential, no sedative effects, no cognitive or psychomotor impairment properties, and no significant withdrawal symptoms, thus representing advantages on the used BDZs [59]. 

In conclusion, all the presented results reinforce the anxiolytic-like effects of palmitone with an EEG profile more similar to BUSP than DZP, likely mediated via multiple neurotransmitter systems at CNS. The effect of palmitone was equivalent to that produced by 5-HT_1A_ receptor agonists but 50% less effective than DZP. The presence of PTX and WAY prevented the anxiolytic-like response of DZP and 8-OH-DPAT, respectively. Whereas only the antagonist of the 5-HT_1A_ receptor (WAY) inhibited the palmitone effects. Palmitone and BUSP exhibited similar changes in the relative power bands after the spectral power analysis, and this response was different from those induced by DZP.

## 4. Materials and Methods

### 4.1. Animals

Adult male Swiss albino mice (28–35 g, 60–70-days old) were used in groups of at least six individuals. They were housed in a room under standard conditions of temperature (22 ± 1 °C) and a humidity-controlled environment on a 12:12 h light–dark cycle (light from 07:00 to 19:00). Animals had free access to water and food (Rodent diet 5001, Lab Diet, USA). All the experiments were conducted under a protocol per the national technical guidelines for the production, care, and use of laboratory animals issued by SAGARPA México (NOM-062-ZOO-1999) and approved by the specifications given by the Ethics and Research committees of the Instituto Nacional de Psiquiatría Ramón de la Fuente Muñiz (CONBIOETICA-09-CEI-010-20170316 and NC093280.1, respectively).

### 4.2. Reagents and Drugs

Diazepam (DZP) (Psicopharma^®^, Ciudad de México, Mexico), buspirone (BUSP), (±)-8-hydroxy-2-(dipropylamine) tetralin hydrobromide (8-OH-DPAT), WAY100635 (WAY), and picrotoxin (PTX) were purchased from Sigma-Aldrich (St. Louis, MO, USA). Palmitone (Figure 12) was obtained and characterized by previous bioguided fractionation from *Annona diversifolia* Saff. Leaves [21]. Drugs were freshly prepared on the day of the experiments and administered intraperitoneal (i.p.) or subcutaneous (s.c.) route using a 0.1 mL/10 g body weight volume. BUSP, 8-OH-DPAT, WAY, and PTX were dissolved in saline solution (s.s., 0.9% NaCl), whereas DZP and palmitone were dissolved in tween 80 (0.2–0.5% in s.s).

### 4.3. Experimental Design

Doses were selected according to the literature and from previous reports of our group [21,27]; PTX [60]. Seventy-two mice were divided into 11 groups of six animals each to receive a single dose of one of the following treatments: 

In the first experimental design, the behavioral response and the EEG profile of mice treated with palmitone, or the anxiolytic drugs alone were explored: -Group 1 was injected with vehicle (0.5% tween 80 in s.s.).-Group 2 was administered palmitone (30 mg/kg, i.p.).-Groups 3 and 4 were injected with DZP (1 mg/kg, i.p.) or BUSP (4 mg/kg, i.p.), respectively. 

In the second experimental design, the role of serotonin 5-HT_1A_ and GABA_A_ receptors was investigated in the presence of palmitone or the selective agonists when combined with their corresponding antagonists as follows:-Group 5 received DZP (GABA_A_ agonist, 1 mg/kg, i.p.) plus PTX (blocker of GABA_A_ receptor chloride channel, 1 mg/kg, i.p.).-Groups 6 and 7 received 8-OH-DPAT (5-HT_1A_ agonist, 1 mg/kg, i.p.) alone, or 8-OH-DPAT plus WAY (5-HT_1A_ antagonist, 1 mg/kg, i.p.), respectively.-Groups 8 and 9 were administered with each antagonist alone: PTX (1 mg/kg, i.p.) or WAY (1 mg/kg, i.p.), respectively. -Groups 10 and 11 received a combination of palmitone (30 mg/kg, i.p.) plus PTX (1 mg/kg, i.p.) or palmitone plus WAY.

Vehicles or antagonists of the GABA_A_ (PTX) and serotonin 5-HT_1A_ (WAY) receptors were injected after a basal EEG recording of 10 min. Fifteen minutes later, treatment with palmitone or with GABA_A_ (DZP) or serotonin 5-HT_1A_ (8-OH-DPAT) receptors agonists was administered. Then, anxiety-like behavior and EEG were registered for 30 min. 

### 4.4. Anxiety Behavioral Tests

#### 4.4.1. Open-Field

Thirty minutes post-administration of treatments, each mouse was individually placed in a 12-quadrant divided acrylic cage (4 cm × 4 cm) to allow mice a free exploration for 2 min. A significant decrease in the number of explored squares by mice receiving treatment compared to those receiving the vehicle was considered an anxiolytic-like response [61].

#### 4.4.2. Hole-Board 

Each mouse was individually placed into an acrylic cage (30 cm in height and 25 cm in depth) to allow mice a free exploration for 3 min. The instrument consisted of a wooden floor (40 cm × 40 cm) with 12 holes 3 cm in diameter evenly distributed. The number of holes explored by mice was immediately counted when the animal introduced its complete head into a hole. A significant reduction in the number of holes explored by mice receiving treatment compared to those in the vehicle group was considered an anxiolytic-like response [62].

#### 4.4.3. Plus-Maze

This test was carried out using an instrument consisting of four wooden arms: two open arms (25 cm long × 5 cm wide) and two closed (25 cm long × 5 cm wide and 15 cm walls) joined by a central part and raised 50 cm from the ground. Each mouse was placed in the center of the cross, and the time each mouse spent in the open or closed arms was recorded as the anxiolytic-like response, as well as the number of crosses in each arm during a period of 5 min [63].

### 4.5. Surgery for EEG and Recordings

#### 4.5.1. Electrode Implantation

Animals were anesthetized with a mixture of xylazine (10 mg/kg, i.p.)/ketamine (100 mg/kg, i.p.) and placed in a stereotaxic mouse frame for electrode implantation (Stoelting, Wood Dale, IL, USA). Four stainless steel screws (1 mm diameter) were screwed bilaterally into the skull over the frontal (AP + 2.0 mm; ML 1 mm) and parietal cortex (AP −3 mm; ML 2.5 mm) [64]. An additional screw was implanted over the cerebellum to serve as a reference (AP −5 mm; ML 0). All the electrodes were soldered to a connector and secured on the skull with dental acrylic. Animals were singly housed after surgery and allowed to recover for at least one week before being submitted to experiments.

#### 4.5.2. EEG Recordings 

On the day of the experiments, animals were placed in a soundproof chamber for 20 min for habituation to the recording conditions. During the experiments, EEG activity was recorded for 40 min: 10 min of baseline activity and 30 min after treatment administration (see timeline in Figure 13A). In the case of groups receiving antagonists such as WAY (1 mg/kg, i.p.) or PTX (1 mg/kg, i.p.), total EEG recording was 55 min: 10 min of baseline activity, 15 min after administration of WAY (1 mg/kg, i.p.) or PTX (1 mg/kg, i.p.), and 30 min after administration of palmitone (30 mg/kg, i.p.) or 8-OH-DPAT (1 mg/kg, s.c.). Animals treated with 8-OH-DPAT (1 mg/kg, s.c.) or DZP received vehicle 15 min before i.p. injection (see timeline in Figure 13B). The behavioral activity of the mice was videotaped simultaneously with the EEG recording. EEG signals from the frontal and parietal cortex were amplified and band-pass filtered (1–70 Hz) using a GRASS model 8-18D electroencephalograph (GRASS Instrument Co., Quincy, MA, USA) and digitized at 500 Hz using an ADQCH8 digital/analog acquisition system [65]. 

#### 4.5.3. EEG Spectral Power Analysis 

The spectral power analysis was performed offline in the EEG activity of both the frontal and parietal cortex using MATLAB (v. 2016a, The Mathworks Inc., Natick, MA, USA). The spectral power was calculated from five 60-s EEG segments of each experimental condition: baseline and 25–30 min after treatments. For the spectral power analysis, baseline segments were considered in which animals remained awake accompanied by a high-frequency/low-amplitude cortical EEG activity. The EEG segments were analyzed by the fast Fourier transform using the fft.m function (0.1 Hz of spectral resolution, 50% of Hamming window overlap) (The Mathworks Inc.) in a custom-made routine. The relative power was obtained by dividing the power value of each frequency by the sum of the frequencies, as previously reported [27]. The relative power spectra were calculated in the band from 1 to 50 Hz and divided into the following frequency bands: delta (1–4 Hz), theta (4–8 Hz), alpha (8–13 Hz), beta (13–30 Hz), and gamma (30–50 Hz). Then, the relative power values obtained from segments of the same experimental condition (i.e., baseline activity or 25–30 min after treatment) were averaged in each frequency band. The change in the activity of the cortical EEG after each experimental treatment was evaluated by normalizing the relative power values for each frequency band as percentage (%) changes of the baseline values ([after treatment period − baseline]/baseline × 100).

### 4.6. Docking Analysis

To support a possible interaction of palmitone with the serotonin 5-HT_1A_ or GABA_A_ receptors (Figure 8A and Figure 8B, respectively), the crystalline structure of the compound was obtained from the PubChem database. This structure was protonated using the Avogadro software V1.2.0 (Avogadro Chemistry, Pittsburgh, PA, USA) for a pH of 7.4. Subsequently, the minimum-energy spatial configuration of palmitone was determined using the Merck Molecular Force Field (MMFF94, Merck Research Laboratories, Boston, MA, USA). The protein structure of the serotonin 5-HT_1A_ or GABA_A_ receptors was obtained from the Protein Data Bank (PDB, https://www.rcsb.org/ accessed on 22 January 2023) with a resolution of ≤3.3 Å. The docking analysis was carried out with the CB-Dock2 tool [66], and the results from the CB-Dock tool software V2.0 (Structural Bio-informatics Research group, Chengdu, China) were contrasted with UCSF Chimera 1.16 (Resource for Biocomputing, Visualization, and Informatics University of California, San Francisco, CA, USA) for protein preparation and AutoDock Vina 1.1.2 (Oleg Trott, La Jolla, CA, USA). A similar exploration was performed with agonists from these two receptors, such as BUSP and 8-OH-DPAT, as well as DZP, to describe their interactions compared to palmitone (Figure 9, Figure 10 and Figure 11, respectively). 

### 4.7. Statistical Analysis

Data are presented as the mean ± standard error of the mean (S.E.M.). The behavioral responses were analyzed by a one-way analysis of variance (ANOVA) followed by Tukey’s *post hoc* test compared to control or between treatments, respectively, of at least six repetitions. The effects of each treatment on the relative power bands compared to baseline values were analyzed by one-sample *t*-tests (testing against baseline values normalized to 0) to assess the direction of the changes (increase or decrease) [67,68]. The comparison of the relative power between treatments was analyzed by a one-way ANOVA followed by Tukey’s *post hoc* test. The *p* values lower than 5% (*p* < 0.5) were considered statistically significant. All the statistical analysis was performed using the SPSS statistical package (V20.0, IBM Corporation, New York, NY, USA) or GraphPad Prism (v8; GraphPad Software Inc., La Jolla, CA, USA).

## Figures and Tables

**Figure 1 molecules-28-03680-f001:**
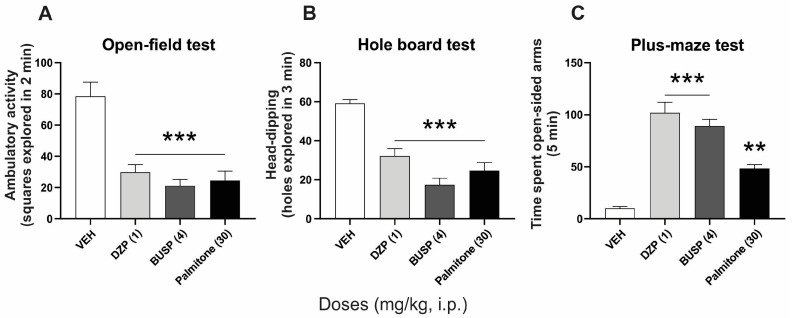
Anxiolytic-like effects of palmitone (30 mg/kg, i.p.), diazepam (DZP, 1 mg/kg, i.p.), and buspirone (BUSP, 4 mg/kg, i.p.) in comparison to the vehicle group (VEH) evaluated in (**A**) the open-field, (**B**) hole-board, and (**C**) plus-maze tests in mice, n = 6 per group. Data are shown as the mean ± S.E.M of six repetitions. One-way ANOVA followed by Tukey’s *post hoc* test, ** *p* < 0.001, *** *p* < 0.0001 vs. VEH.

**Figure 2 molecules-28-03680-f002:**
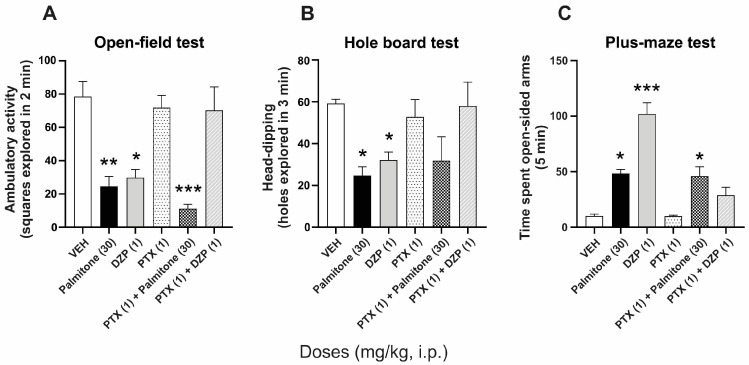
Anxiolytic-like effects of palmitone (30 mg/kg, i.p.) and DZP (1 mg/kg, i.p.) alone and in the presence of picrotoxin (PTX, 1 mg/kg, i.p.) in (**A**) the open-field, (**B**) hole-board, and (**C**) plus-maze tests in mice. Data are shown as the mean ± S.E.M. of six repetitions. One-way ANOVA followed by Tukey’s *post hoc* test, * *p* < 0.005, ** *p* < 0.001, *** *p* < 0.0001 vs. vehicle (VEH).

**Figure 3 molecules-28-03680-f003:**
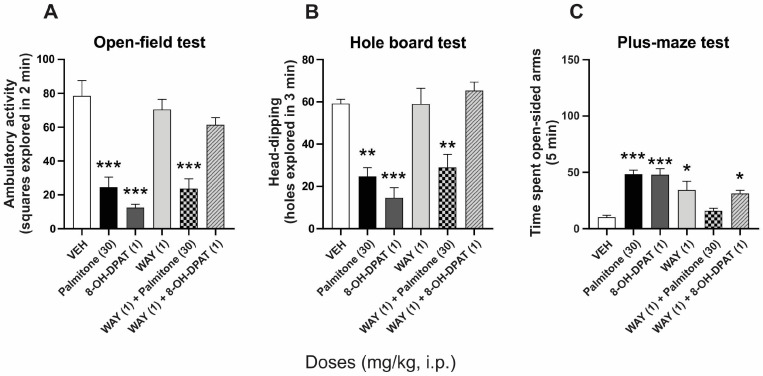
Anxiolytic-like effects of palmitone (30 mg/kg, i.p.) and 8-OH-DPAT (1 mg/kg, s.c.) alone and in the presence of WAY100635 (WAY, 1 mg/kg, i.p.) in (**A**) the open-field, (**B**) hole-board, and (**C**) plus-maze tests in mice. One-way ANOVA followed by Tukey’s *post hoc* test, * *p* < 0.005, ** *p* < 0.001, *** *p* < 0.0001 vs. vehicle (VEH).

**Figure 4 molecules-28-03680-f004:**
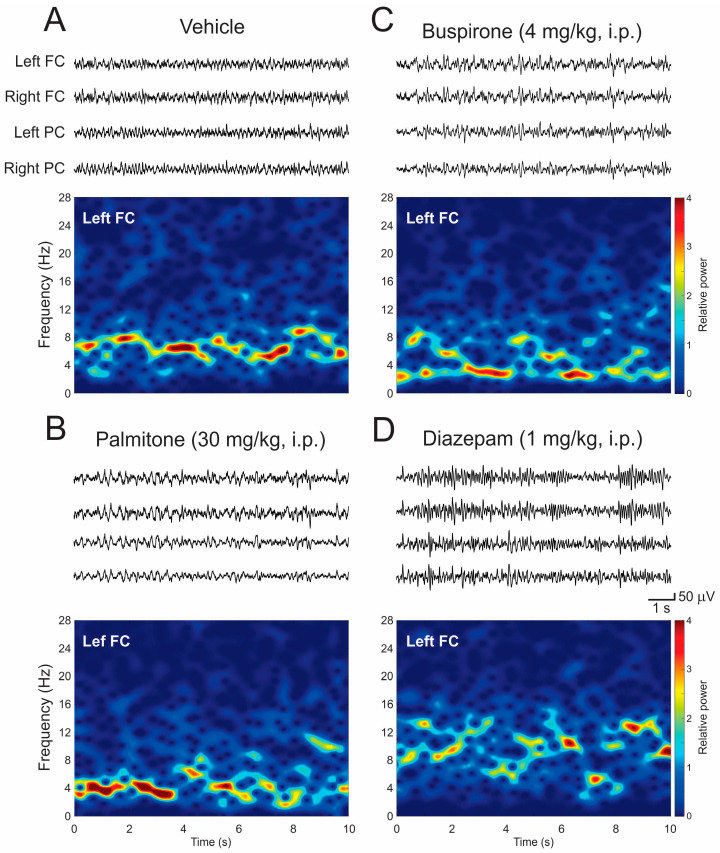
Representative EEG patterns following administration of treatments. (**A**–**D**) Representative EEG recordings from the frontal and parietal cortex, thirty minutes after treatments, and relative power spectra from Left FC in lower. Abbreviations: EEG, electroencephalogram; Left FC, left frontal cortex; Right FC, right frontal cortex; Left PC, left parietal cortex; Right PC, right parietal cortex.

**Figure 5 molecules-28-03680-f005:**
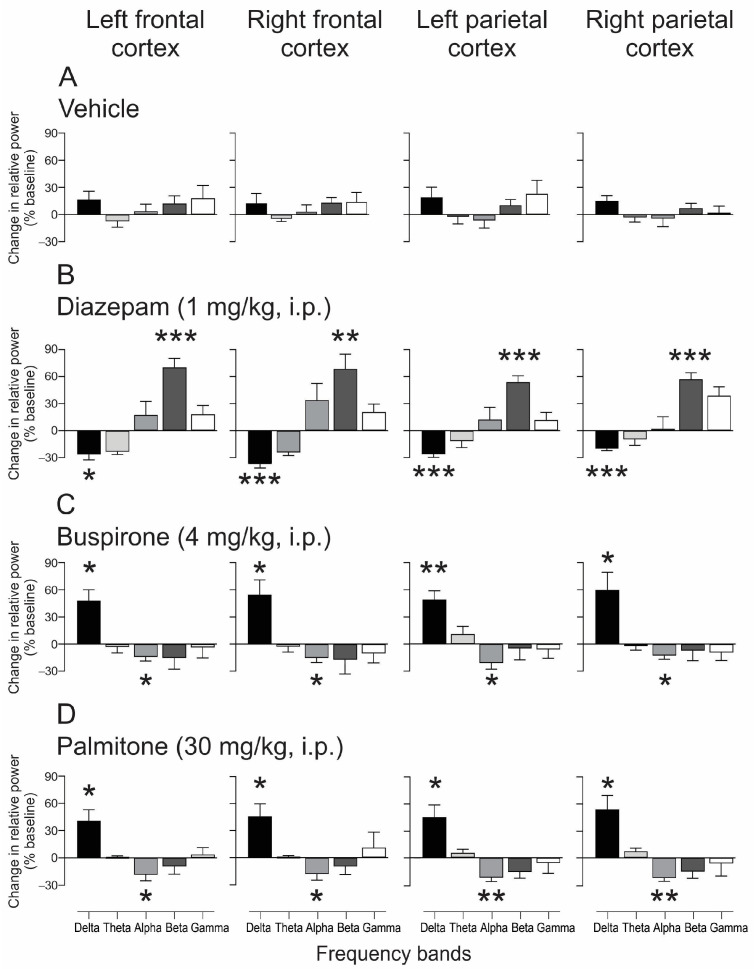
Effects of the anxiolytic treatments on the relative power bands in mice. Normalized relative power (% from baseline) at 30 min post-treatments on the right frontal and parietal cortex. Each treatment provoked different changes in the cortical activity spectral profile. Notice a decrease in the delta band and an increase in the beta band after diazepam (1 mg/kg, i.p.) in the frontal and parietal cortex (**B**). Regarding buspirone (4 mg/kg, i.p.) and palmitone (30 mg/kg, i.p.), they provoked increases in the delta and decreases in the alpha bands in the frontal and parietal cortex ((**C**,**D**), respectively). The vehicle did not show statistical differences (**A**). Data are shown as the mean ± S.E.M. of six repetitions. One-sample *t*-tests, * *p* < 0.05, ** *p* < 0.005, *** *p* < 0.001 vs. baseline.

**Figure 6 molecules-28-03680-f006:**
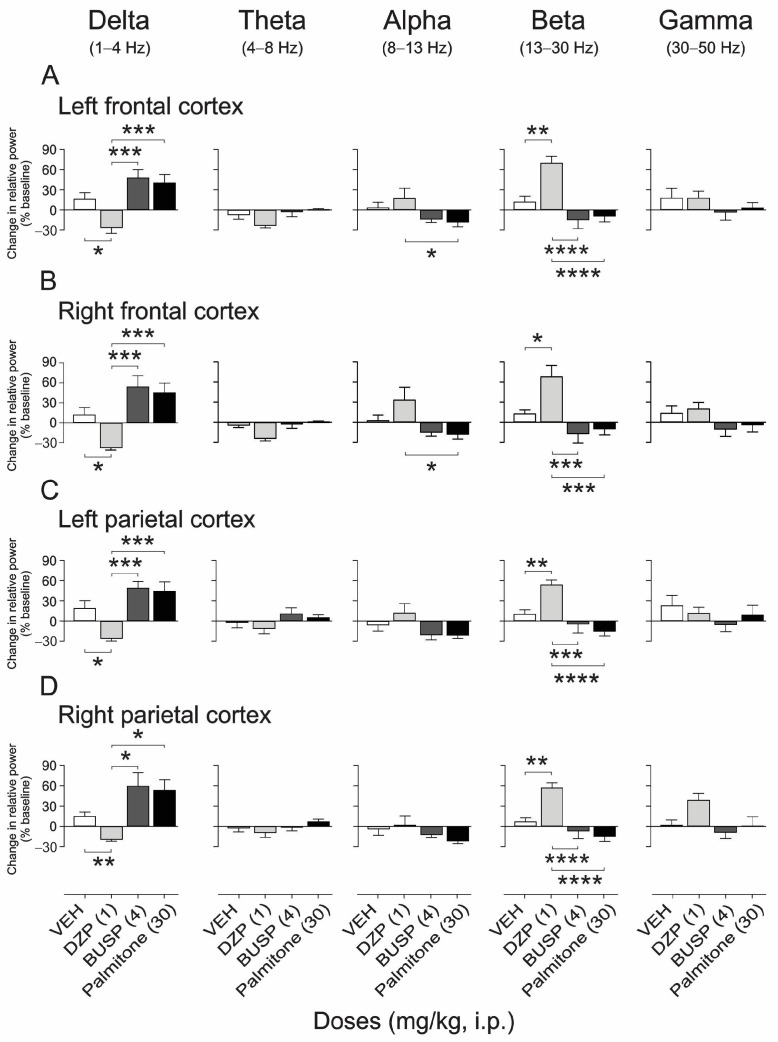
Comparison among experimental groups on the relative power bands. Palmitone (30 mg/kg, i.p.) and buspirone (BUSP 4 mg/kg, i.p.) showed a significant increase in the delta band in the frontal and parietal cortex compared to diazepam (DZP 1 mg/kg, i.p.) ((**A**–**D**) left panels). In contrast, DZP showed a higher proportion of the beta frequency band in comparison to palmitone, BUSP, and vehicle (VEH) ((**A**–**D**) right panels), and a significant diminution of the delta band compared to VEH ((**A**–**D**) left panels) in the four analyzed regions. There were no statistical differences in the theta, alpha, and gamma frequency bands among groups. Data are shown as the mean ± S.E.M. of six repetitions. One-way ANOVA followed by Tukey’s *post hoc* test, * *p* < 0.05, ** *p* < 0.01, *** *p* < 0.001, **** *p* < 0.0001 vs. DZP.

**Figure 7 molecules-28-03680-f007:**
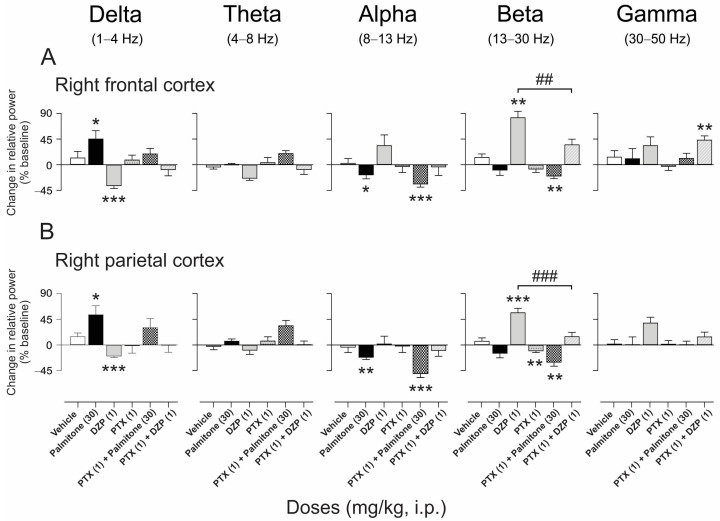
Effects of palmitone (30 mg/kg, i.p.) and DZP (1 mg/kg, i.p.) in the presence of the GABA_A_ antagonist picrotoxin (PTX, 1 mg/kg, i.p.) on the relative power bands of the right frontal (**A**) and parietal cortex (**B**). Data are shown as the mean ± S.E.M of six repetitions per group. One-sample *t*-tests, * *p* < 0.05, ** *p* < 0.01, *** *p* < 0.005 vs. baseline. One-way ANOVA followed by Tukey’s *post hoc* test, ## *p* < 0.001, ### *p* < 0.0001 DZP vs. PTX + DZP.

**Figure 8 molecules-28-03680-f008:**
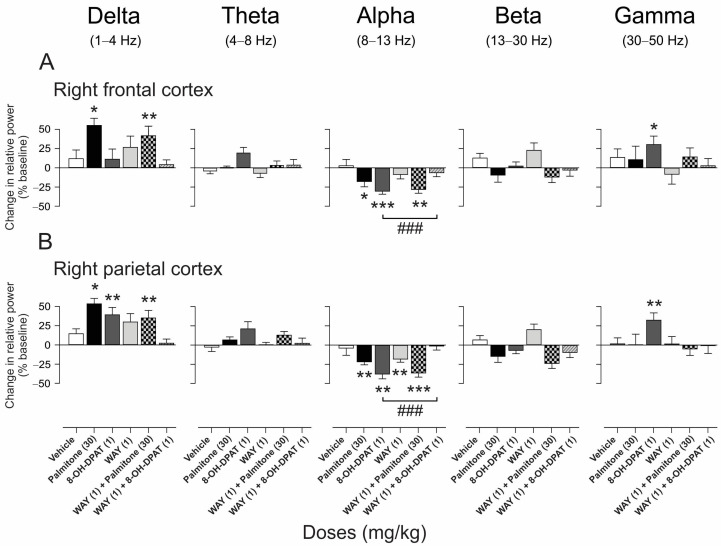
Effects of palmitone (30 mg/kg, i.p.) and 8-OH-DPAT (1 mg/kg, s.c.) in the presence of the serotonin 5-HT_1A_ receptor antagonist WAY100635 (WAY, 1 mg/kg, i.p.) on the relative power bands of the right frontal (**A**) and parietal cortex (**B**). Data are shown as the mean ± S.E.M of six repetitions per group. One-sample *t*-tests, * *p* < 0.05, ** *p* < 0.01, *** *p* < 0.005 vs. baseline. One-way ANOVA followed by Tukey’s *post hoc* test, ### *p* < 0.0001 8-OH-DPAT vs. WAY + 8-OH-DPAT.

**Figure 9 molecules-28-03680-f009:**
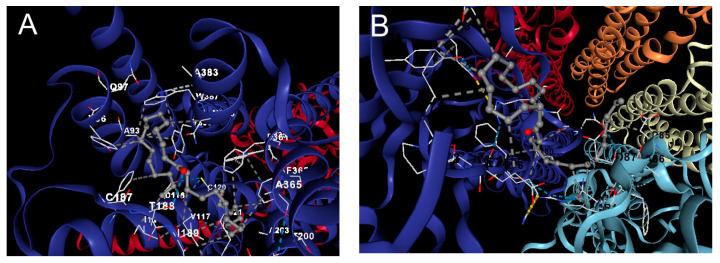
In silico evaluation of the interaction of palmitone on the serotonin 5-HT_1A_ receptor (**A**) and GABA_A_ receptor (**B**). The first complex shared 27 steric interactions, chain R: ALA93, **TYR96**, GLN97, **PHE112**, **ILE113**, **ASP116**, **VAL117**, **CYS120**, THR121, **CYS187**, **THR188**, **ILE189**, **SER190**, **LYS191**, **TYR195**, **THR196**, **SER199**, **THR200**, ALA203, TRP358, **PHE361**, **PHE362**, **ALA365**, ALA383, ASN386, TRP387, TYR390 (**A**). While palmitone and GABA_A_ complex shared 22 steric interactions, such as chain D: ASN39, ASP85, ALA86, ASP87, ARG104, PHE105, SER106, and chain E: PRO67, **ILE72**, **PRO73**, **GLU74**, ILE75, **ARG76**, PHE77, VAL80, ASN82, ALA83, ARG84, ALA86, VAL88, TYR101, GLU103 (**B**). Amino acids labeled in bold shared similar interaction with BUSP.

**Figure 10 molecules-28-03680-f010:**
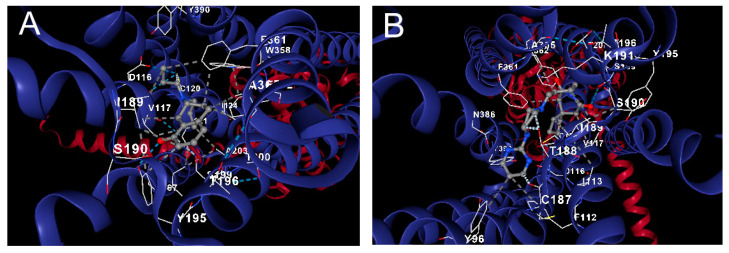
In silico evaluation of the interaction of buspirone (BUSP) and 8-OH-DPAT on the serotonin 5-HT_1A_ receptor (**A** and **B**, respectively). BUSP and 5-HT_1A_ complex shared 19 steric interactions, chain R: ASP116, VAL117, CYS120, THR121, ILE124, ILE167, ILE189, SER190, TYR195, THR196, SER199, THR200, ALA203, TRP358, PHE361, PHE362, ALA365, ASN386, TYR390 (**A**). While, 21 steric interactions were obtained for 8-OH-DPAT, chain R: TYR96, PHE112, ILE113, ASP116, VAL117, CYS120, CYS187, THR188, ILE189, SER190, LYS191, TYR195, THR196, SER199, THR200, PHE361, PHE362, ALA365, ASN386, TYR390.

**Figure 11 molecules-28-03680-f011:**
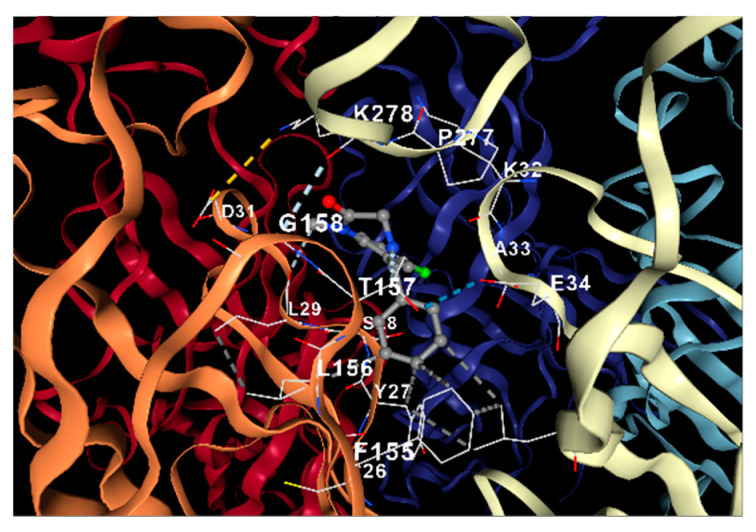
Molecular docking of DZP on the GABA_A_ receptor showed chain D: THR64, TYR65, GLU66, VAL89. Chain E: VAL4, SER5, TRP71, ILE72, PRO73, GLU74, ARG76, VAL131, ARG132, VAL134, THR136, ILE139.

**Figure 12 molecules-28-03680-f012:**
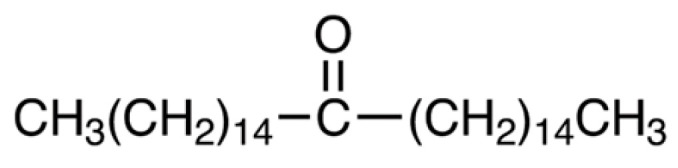
Palmitone (16-hentriacontanone).

**Figure 13 molecules-28-03680-f013:**
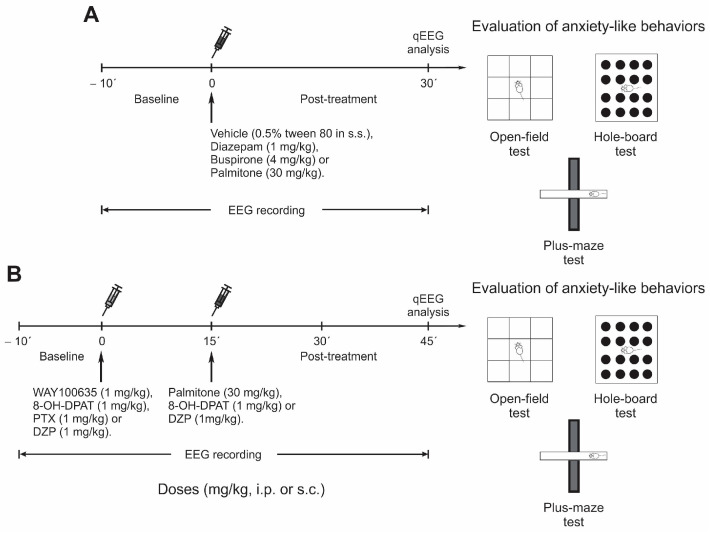
Timeline of experimental treatments. (**A**) Animals were treated with vehicle (0.5% tween 80 in 0.9% s.s., i.p.), palmitone (30 mg/kg, i.p.), buspirone (BUSP, 4 mg/kg, i.p.) or diazepam (DZP, 1 mg/kg, i.p.) 30 min before behavioral tests (open-field, hole-board, and elevated plus-maze). (**B**) To investigate the involvement of the serotonin 5HT_1A_ receptors in the anxiolytic-like effect induced by palmitone, the antagonist WAY100635 (WAY, 1 mg/kg, i.p.) or the agonist 8-OH-DPAT (1 mg/kg, s.c.) were administered 15 min before palmitone (30 mg/kg, i.p.) or 8-OH-DPAT (1 mg/kg, s.c.). In addition, the GABA_A_ receptor antagonist picrotoxin (PTX, 1 mg/kg, i.p.) was administered alone or in combination with palmitone (30 mg/kg, i.p.). Thirty minutes after the administration of the treatments, behavioral tests were performed. In all experiments, EEG activity was recorded up to behavioral tests. Abbreviations: electroencephalographic (EEG) recording; intraperitoneal injection (i.p.); quantitative electroencephalogram analysis (qEEG); subcutaneous injection (s.c.).

## Data Availability

The data presented in this study are available on request from the corresponding authors due to privacy or ethical restrictions.

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
