# Peer review of "Anxiolytic-like Effects and Quantitative EEG Profile of Palmitone Induces Responses Like Buspirone Rather Than Diazepam as Clinical Drugs"

_molecules, 2023, doi:10.3390/molecules28093680_

Round 1

Reviewer 1 Report

Reviewer’s Comments:

The manuscript “Quantitative EEG profile of the anxiolytic-like effects of palmitone in comparison to clinical drugs diazepam and buspirone in mice” is a very interesting work. In this work, anxiety is a mental disorder with a growing worldwide incidence due to the SARS-CoV-2 virus pandemic. Pharmacological therapy includes drugs such as benzodiazepines (BDZs) or azapirones like buspirone (BUSP) or analogs, which unfortunately produce severe adverse effects or no immediate response, respectively. Medicinal plants or their bioactive metabolites are a shared global alternative to treat anxiety. Palmitone is one active compound isolated from Annona species due to its tranquilizing activity. However, its influence on neural activity and possible mechanism of action are unknown. In this study, an electroencephalographic (EEG) spectral power analysis was used to corroborate its depressant activity in comparison with the anxiolytic-like effects of reference drugs such as diazepam (DZP, 1 mg/kg) and BUSP (4 mg/kg) or 8-OH-DPAT (1 mg/kg), alone or in the presence of the GABAA (picrotoxin, PTX, 1 mg/kg) or serotonin 5-HT1A receptor antagonists (WAY100634, WAY, 1 mg/kg). While I believe this topic is of great interest to our readers, I think it needs major revision before it is ready for publication. So, I recommend this manuscript for publication with major revisions.

1. In this manuscript, the authors did not explain the importance of the palmitone in the introduction part. The authors should explain the importance of palmitone.

2) Title: The title of the manuscript is not impressive. It should be modified or rewritten it.

3) Correct the following statement “The presence of PTX and WAY prevented the response of DZP and 8-OH-DPAT, respectively. However, only WAY modified the effect of palmitone. In the spectral power analysis, palmitone and BUSP exhibited similar changes in the relative power bands that contrast the changes induced by DZP. In conclusion, brain electrical activity was associated with palmitone’s anxiolytic-like effects that implicate a serotoninergic rather than a GABAergic mechanism of action”.

4) Keywords: palmitone is missing in the keywords. So, modify the keywords.

5) Introduction part is not impressive. The references cited are very old. So, Improve it with some latest literature like 10.3390/molecules27217368, 10.3390/pr10081455

6) The authors should explain the following statement with recent references, “Thus, data obtained from the right frontal and parietal cortex from EEG are reported in the main text (the EEG analysis dataset from the left hemisphere can be found in Supplementary Figures S1 and S2)”.

7) Add space between magnitude and unit. For example, in synthesis “21.96g” should be 21.96 g. Make the corrections throughout the manuscript regarding values and units.

8) The author should provide reason about this statement “The precise mechanism of palmitone in producing its anxiolytic effects remains unknown”.

9. Comparison of the present results with other similar findings in the literature should be discussed in more detail. This is necessary in order to place this work together with other work in the field and to give more credibility to the present results.

10) Conclusion part is very long. Make it brief and improve by adding the results of your studies.

11) There are many grammatic mistakes. Improve the English grammar of the manuscript.

Author Response

Response to Reviewer 1 Comments:

While I believe this topic is of great interest to our readers, I think it needs major revision before it is ready for publication. So, I recommend this manuscript for publication with major revisions.

Point 1. In this manuscript, the authors did not explain the importance of the palmitone in the introduction part. The authors should explain the importance of palmitone.

Response 1: Thank you for your recommendation. Information emphasizing the palmitone relevance was included in a paragraph of the introduction section.

Palmitone (16-hentriacontanone) has been used to produce lipidized nucleosides as potential brain markers with the ability to cross the blood brain barrier as a Single Photon Emission Computed Tomography (SPECT) agent for neuroimaging studies to improve early diagnostic (Swastika et al., 2019) supporting its easy pass through the brain tissue. In the brain, palmitone produces depressant activity reducing excitability like paroxysms and protecting cerebral regions such as hippocampus from neuronal damage, not only in adult rats but also in prepubertal rats (González-Trujano et al., 2006; Cano-Europa et al., 2010). Concerning pain relief, palmitone reduced nociceptive response in abdominal, nociceptive, and inflammatory tests in rodents involving opioid and 5-HT1A receptors (Carballo et al., 2010). It has been found to be a major component representing up to 48% of the total cuticular wax of A. squamosa which antimicrobial activity against selected Gram-positive and Gram-negative bacterial and fungal strains have been reported to be significantly higher or equivalent in the presence of palmitone, respectively, in comparison to some isomeric hydroxy ketones (Shanker et al., 2007). All these studies together support the biological relevance of palmitone on the CNS and in other cases.

Point 2: Title: The title of the manuscript is not impressive. It should be modified or rewritten it.

Response 2: Thank you for your suggestion, a modified title was included.

Anxiolytic-like effects and quantitative EEG profile of palmitone induced responses like buspirone rather than diazepam as clinical drugs

Point 3: Correct the following statement “The presence of PTX and WAY prevented the response of DZP and 8-OH-DPAT, respectively. However, only WAY modified the effect of palmitone. In the spectral power analysis, palmitone and BUSP exhibited similar changes in the relative power bands that contrast the changes induced by DZP. In conclusion, brain electrical activity was associated with palmitone’s anxiolytic-like effects that implicate a serotoninergic rather than a GABAergic mechanism of action”.

Response 3: Thank you for your recommendation. This phase was corrected and precise as follows:

“The presence of PTX and WAY prevented the anxiolytic-like response of DZP and 8-OH-DPAT, respectively. Whereas only the antagonist of the 5-HT1A receptor (WAY) inhibited the palmitone effects. On the other hand, palmitone and BUSP exhibited similar changes in the relative power bands after the spectral power analysis. This response was different to the changes induced by DZP. In conclusion, brain electrical activity was associated with the anxiolytic effects of palmitone implying a serotonergic rather than a GABAergic mechanism of action."

Point 4: Keywords: palmitone is missing in the keywords. So, modify the keywords.

Response 4: Thank you for the observation. Palmitone was included in the keywords.

Point 5: Introduction part is not impressive. The references cited are very old. So, Improve it with some latest literature like 10.3390/molecules27217368, 10.3390/pr10081455

Response 5: Thank you for your observation. We have updated our bibliographic research to select relevant references supporting our present study.

Point 6: The authors should explain the following statement with recent references, “Thus, data obtained from the right frontal and parietal cortex from EEG are reported in the main text (the EEG analysis dataset from the left hemisphere can be found in Supplementary Figures S1 and S2)”.

Response 6: Thank you for your comment. The mistake in the text was corrected. In addition, some references supporting our results were added as follows:

In order to not duplicate results, data obtained from the right frontal and parietal cortex from EEG were reported in the main text, whereas the EEG analysis dataset from the left hemisphere can be found in Supplementary Figures S1 and S2.

We included a paragraph into the discussion section:

These results agree with the human and rodent literature indicating that the effect of drugs such as BDZs on the brain electric activity is equivalent in central and parietal regions (Christian et al., 2015; Manor et al., 2021; Nishida et al., 2016; Victorino et al., 2021), and emphasize the importance of multi-lead recording in quantitative analysis of the EEG in order to document the effects on brain electric activity in the presence of drugs or natural products.

Point 7: Add space between magnitude and unit. For example, in synthesis “21.96g” should be 21.96 g. Make the corrections throughout the manuscript regarding values and units.

Response 7: Our apologies for mistakes. Document was checked and corrected all along the text.

Point 8: The author should provide reason about this statement “The precise mechanism of palmitone in producing its anxiolytic effects remains unknown”.

Response 8: Thank you for your comment. Although the depressant activity of this molecule has already been reported as an anticonvulsant and anxiolytic with a possible GABAergic mechanism, there is no description for other specific receptors or targets. One reason for these lines was explained in the text as follows:

Preliminary studies reporting biological effects of isolated palmitone from the hexane extracts obtained from Annona diversifolia and A. cherimolia on the CNS, such as anticonvulsant and anxiolytic (González-Trujano et al., 2001;2006; López-Rubalcava et al., 2006) suggested a possible mechanism of action through GABAergic neurotransmission. Another study reported that opioids and serotonin 5-HT1A receptors were involved in the antinociceptive effects of A. diversifolia (Carballo et al., 2006). However, direct exploration of this pure compound alone and in the presence of selective antagonists was not previously investigated for the antianxiety response. To clarify a possible mechanism of action for palmitone in preclinical anxiety, the participation of serotonin 5-HT1A receptor but not the GABA/BDZs site receptor was corroborated in this study by using selective antagonists alone or combined with palmitone. Our present results using not only behavior but also EEG analysis reinforced that this aliphatic ketone differed from tht observed for BDZ such as diazepam.

Point 9: Comparison of the present results with other similar findings in the literature should be discussed in more detail. This is necessary in order to place this work together with other work in the field and to give more credibility to the present results.

Response 9: Thank you for your comment. We have considered your suggestion when comparing equivalent results with other similar molecules to strengthen our data description in a more emphatic manner complementing information already included in the text.

The presence of palmitone as abundant secondary metabolite has been found in several plants such as in the hexane extract of leaves of Aristolochia cordigera (Pereira et al., 2017) or in the Annona species (González-Trujano et al., 2006; López-Rubalcava et al., 2006), but also in the ethanol extracts (González-Trujano et al., 2001). For example, the anxiolytic-like effects of a methanol extract of the Aristolochia indica leaves was reported using light/dark test (Pooja et al., 2015). Several fatty acids, such as palmitoleic acid (0.14%), palmitic acid (21.42%), linoleic acid (11.02%), oleic acid (59.97%), and stearic acid (7.44%), were identified as constituents of an hexanic extract of the Bertholletia excelsa seeds producing significant anxiolytic-like effects (Frausto-González et al., 2021).

Point 10: Conclusion part is very long. Make it brief and improve by adding the results of your studies.

Response 10: Thanks for your suggestion. The conclusion was modified to include the most important results.

In conclusion, all the presented results reinforce the anxiolytic-like effects of palmitone with an EEG profile more similar to BUSP than DZP, likely mediated via multiple neurotransmitter systems at CNS. The effect of palmitone was equivalent to that produced by 5-HT1A receptor agonists but 50% less effective than DZP. The presence of PTX and WAY prevented the anxiolytic-like response of DZP and 8-OH-DPAT, respectively. Whereas only the antagonist of the 5-HT1A receptor (WAY) inhibited the palmitone effects. On the other hand, palmitone and BUSP exhibited similar changes in the relative power bands after the spectral power analysis, but this response was different to the responses induced by DZP.

Point 11: There are many grammatic mistakes. Improve the English grammar of the manuscript.

Response 11: Thank you for your observation. We have again reviewed the entire document for spelling errors.

Reviewer 2 Report

In this study, the authors have investigated the neural activity  of palmitone, an active compound isolated from Annona species, to determine it's potential as an alternate to anti anxiety medicines as benzodiazepines or azapirones like buspirone (BUSP). The anxiolytic-like activity was probed using the behavioral response of mice. The authors used electroencephalographic (EEG) spectral power analysis to measure the depressant activity in comparison with other drugs like  diazepam, BUSP on its own or with picrotoxin or serotonin 5-HT1A receptor antagonists. The paper is well written, clear and thorough. The authors provide important insights to the mechanism of neural activity of palmitone making an interesting case of it as an alternative. 

Since most people with anxiety disorder use the drugs very often, sometimes daily, I have some questions about the effect of palmitone with time. Do the authors know the how the effect of palmitone changes with time compared to the existing drugs? How long do the anxiolytic-like effects last? Is there any information of what are the effects after multiple doses? 

Author Response

Response to Reviewer 2 Comments:

In this study, the authors have investigated the neural activity of palmitone, an active compound isolated from Annona species, to determine it's potential as an alternate to antianxiety medicines as benzodiazepines or azapirones like buspirone (BUSP). The anxiolytic-like activity was probed using the behavioral response of mice. The authors used electroencephalographic (EEG) spectral power analysis to measure the depressant activity in comparison with other drugs like diazepam, BUSP on its own or with picrotoxin or serotonin 5-HT1A receptor antagonists. The paper is well written, clear and thorough. The authors provide important insights to the mechanism of neural activity of palmitone making an interesting case of it as an alternative.

Point 1: Since most people with anxiety disorder use the drugs very often, sometimes daily, I have some questions about the effect of palmitone with time. Do the authors know the how the effect of palmitone changes with time compared to the existing drugs? How long do the anxiolytic-like effects last? Is there any information of what are the effects after multiple doses?

Response 1: Thank you for your review and comments. About your question, in this preliminary investigation, we have explored the anxiolytic-like response of palmitone after acute administration. However, the anticonvulsant effects of palmitone have been explored in our laboratory after repeated administration by using the chemical kindling in mice without observing toxicity (Data not reported yet) and in the electrical amygdala kindling in rats (González-Trujano et al., 2006). This explanation was included in the text as an antecedent of the importance of this kind of molecules and to emphasize that chronic administration is required in the future to explore therapeutic responses for chronic and degenerative diseases as follows:

Therapeutic responses for neurodegenerative diseases require repetitive administration of clinical drugs. Since most people with anxiety disorders use clinical drugs very often, sometimes daily, it is important to consider in future studies the antianxiety effects of palmitone after chronic administration, such as already reported for its anticonvulsant properties in the electrical amygdala kindling in rats without producing toxicity (González-Trujano et al., 2006).

González-Trujano, M. E., López-Meraz, L., Reyes-Ramírez, A., Aguillón, M., & Martínez, A. (2009). Effect of repeated administration of Annona diversifolia Saff. (ilama) extracts and palmitone on rat amygdala kindling. Epilepsy & behavior: E&B, 16(4), 590–595. https://doi.org/10.1016/j.yebeh.2009.09.018
